psychology

adolescence, social preference, social stimuli, non-social stimuli, academic diligence

**Author for correspondence:**
J. L. Andrews
e-mail: jack.andrews.16@ucl.ac.uk

# The effect of social preference on academic diligence in adolescence

J. L. Andrews[1], L. Foulkes[1,2], C. Griffin[1]
and S. J. Blakemore[1]

[1]UCL Institute of Cognitive Neuroscience, 17–19 Queen Square, London WC1N 3AR, UK
[2]Department of Education, University of York, York YO10 5DD, UK

(iD) JLA, 0000-0002-0180-185X; SJB, 0000-0002-1690-2805

In the current study, we were interested in whether adolescents show a preference for social stimuli compared with non-social stimuli in the context of academic diligence, that is, the ability to expend effort on tedious tasks that have long-term benefits. Forty-five female adolescents (aged 11–17) and 46 female adults (aged 23–33) carried out an adapted version of the Academic Diligence Task (ADT). We created two variations of the ADT: a social ADT and non-social ADT. Individuals were required to freely split their time between an easy, boring arithmetic task and looking at a show-reel of photographs of people (in the social ADT) or landscapes (in the non-social ADT). Individuals also provided enjoyment ratings for both the arithmetic task and the set of photographs they viewed. Adolescents reported enjoying the social photographs significantly more than the non-social photographs, with the converse being true for adults. There was no significant difference in the time spent looking at the social photographs between the adolescents and adults. However, adults spent significantly more time than adolescents looking at the non-social photographs, suggesting that adolescents were less motivated to look at the non-social stimuli. Further, the correlation between self-reported enjoyment of the pictures and choice behaviour in the ADT was stronger for adults than for adolescents in the non-social condition, revealing a greater discrepancy between self-reported enjoyment and ADT choice behaviour for adolescents. Our results are discussed within the context of the development of social cognition and introspective awareness between adolescence and adulthood.

## 1. Introduction

Starting with puberty and ending with adult independence, adolescence is a period of significant biological, psychological

and social change [1]. During adolescence, individuals begin to spend more time with their peers, and less time with their parents [2], while also becoming increasingly concerned about the actions and opinions of their peer group [3]. Such changes render adolescence a distinct period of social reorientation [4] and previous research has revealed adolescence to be marked by a heightened effect of peer influence [5,6].

One proposed reason that adolescents are particularly sensitive to social influence is that social stimuli have a higher reward value for adolescents compared to adults [7]. In one study, children (6–12 years old), adolescents (13–17 years old) and adults (18–29 years old) performed a go/no-go task, with both appetitive (happy faces) and neutral (calm faces) cues [8]. Compared with children and adults, adolescents made more errors on no-go appetitive trials (but not on neutral trials). This suggests that, when presented with appetitive cues (happy faces) adolescents exhibit a reduced ability to suppress approach behaviours [8]. In another study, adolescents (12–14 years old) but not adults (18–29 years old) showed reduced performance on an N-Back working memory task when distracted by smiling faces [9]. There is also some evidence that adolescents show hypersensitivity to social reward, as shown by heightened activation in reward-related brain regions such as the ventral striatum [8,10]. However, drawing clear conclusions regarding the reward value of such stimuli from fMRI data alone is problematic given the problem of reverse inference [7,11]. In sum, several studies to date have shown hypersensitivity to positive social stimuli in adolescence, but whether this is due to heightened reward value of social stimuli remains unclear.

Other studies investigating age differences in social reward or social preference have revealed mixed results [7]. For example, social/monetary incentive delay tasks have shown no significant preference for social over non-social stimuli in adolescents for money versus faces [12] nor for images of real people versus cartoons [13]. Furthermore, a preference for non-social versus social stimuli in adolescents has been shown in an approach-avoidance task for cars versus faces [14] and in a visual exploration task in which gaze behaviour was measured across an array of pictures including faces and non-social items related to circumscribed interests (e.g. trains) [15].

In a more recent study, 255 typically developing individuals aged between 4 and 20 years old were evaluated on a social motivation 'Choose a Movie' Task. During this task, participants are asked to choose between viewing social (of people) or non-social (of everyday objects) movies, which are presented with varying levels of effort (key presses required). The typically developing adolescents in this study showed no clear stimuli preference [16]. However, this study did not include participants older than 20 years. Therefore, the current study was developed to explore whether adolescents, compared to adults, show a preference for social versus non-social stimuli.

In order to investigate this question, we used a paradigm designed to study academic diligence, the tendency to expend effort on academic tasks that are momentarily tedious but beneficial in the long term [17]. Academic diligence can be thought of as a domain-specific component of self-control [18]. Exercising self-control in order to stay focused on an academic task, especially given the many distractions adolescents are faced with, is beneficial to future success [18]. Higher academic diligence is related to better academic performance, as well as to other outcomes including good mental health (456 males tracked from 14 to 47 years) [19]. For example, parental reports of their child's diligence (e.g. 'stays at a task until it's done') are correlated with their child's academic outcomes, including high school and college grade point average (GPA) [20].

The most widely used experimental measure of academic diligence is the Academic Diligence Task (ADT) [17]. The ADT was designed to simulate the types of decisions that students are often faced with when doing school work. In this task participants must allocate their time and effort to either a tedious but 'good for you' maths task or a more enjoyable distractor task (such as the video game Tetris™ or watching music videos). In a large sample of adolescents (mean age 17.9 years, s.d. = 0.51), proportion of time spent on the maths predicted GPA, maths and reading scores, and college enrolment more robustly than demographics and IQ [17].

Young people are often faced with a choice between academic work and social activities, such as spending time on social media or with friends in both home and school environments. Thus, in the present study, we adapted the ADT in order to assess whether the opportunity to engage with social or non-social stimuli (photographs) would have a differential effect on academic diligence in adolescents and adults. Participants were presented with basic maths questions and the option to switch out of the maths activity in order to look at a show-reel of photographs. In the social ADT the photographs depicted people, while in the non-social ADT the photographs depicted landscapes. For each version of the task, participants had the option to switch between the basic maths activity and looking at the show-reel of photographs as many times as they liked within a 10 min period. After

each condition, participants also provided enjoyment ratings for the maths questions and the photographs.

The study was designed to test three hypotheses:

(1) There would be a difference in the amount of time adolescents and adults spent looking at photographs in favour of doing maths on the ADT, such that adolescents would be more interested in looking at photographs in the social condition than in the non-social condition and this would not be the case for adults.
(2) Compared with adults, adolescents would report enjoying the social stimuli more than the non-social stimuli.
(3) The time spent looking at the photographs in each condition would positively correlate with the picture enjoyment ratings, for both age groups.

## 2. Method

### 2.1. Participants

Ninety-one female participants, comprising 45 adolescents aged 11–17 (mean age: 15.4; s.d.: 1.81) and 46 adults aged 23–33 years old (mean age: 24.96; s.d.: 2.61), took part in the study. Sample homogeneity was increased by recruiting only female participants, removing variance that could be accounted for by sex differences in puberty onset [21]. Participants were excluded from participating if they had any diagnosed developmental disorder, including dyslexia, dyscalculia or autism spectrum conditions. Participants were recruited from the Greater London area through advertisements in schools, social media and via the Psychology Department subject pool. Participants completed the matrix reasoning subscale of the WASI (Wechsler Abbreviated Scale of Intelligence), and questionnaire measures for autistic traits, social reward and grit (see electronic supplementary material for descriptive information and further discussion of these measures).

The study followed the Research Ethics Guidelines and was approved by the University Research Ethics Committee. Informed consent was obtained from adult participants and consent was obtained from the parent or guardian of participants under 18 years old. Following the completion of the study, participants were debriefed, given the opportunity to ask any questions and compensated £10 for their time. Testing was carried out individually in a quiet room at school or in the laboratory.

### 2.2. Adapted academic diligence task

We developed an adapted version of the Academic Diligence Task (ADT) [17]. As in the original ADT, participants were presented with the option of either completing arithmetic multiple choice problems (e.g. $5 + 5 = ?$; $2 \times 3 = ?$), which were described as 'basic', or engaging with a distractor task. The arithmetic questions were kept simple so that maths ability would not confound diligence, and they were designed to be boring in order to represent the temptation to stop a purportedly beneficial but monotonous task in favour of a more interesting distractor task. Using basic arithmetic problems reduced the likelihood that individuals who would ordinarily enjoy completing maths problems would find the task engaging [22]. This increases the likelihood that any variation in the time spent on the maths is due to diligence.

In the original ADT paradigm, the distractor task gives participants the option of playing games (such as Tetris™) or watching videos. Here, we adapted the ADT such that the distractor task comprised a show-reel of photographs (figure 1). The photographs either depicted people (either alone or interacting; the social ADT), or landscapes (the non-social ADT). The photographs within each condition were presented in a random order, with each picture presented for 5 s before automatically moving to the next photograph. A large selection of social and non-social pictures was collected from online stock picture websites and Google searches. All pictures were free for non-commercial reuse. For the purpose of piloting, the images were rated by six adults for positive valence, and then reduced down to 60 social and 60 non-social pictures. Each photograph was included in both colour and black and white; therefore, a total of 120 social pictures and 120 non-social pictures were included for possible viewing in each condition. The social and the non-social ADT each lasted 10 min. Participants completed both the social and non-social ADT, so the total task duration was 20 min per participant. Condition order (social or non-social ADT first) was counterbalanced across participants.

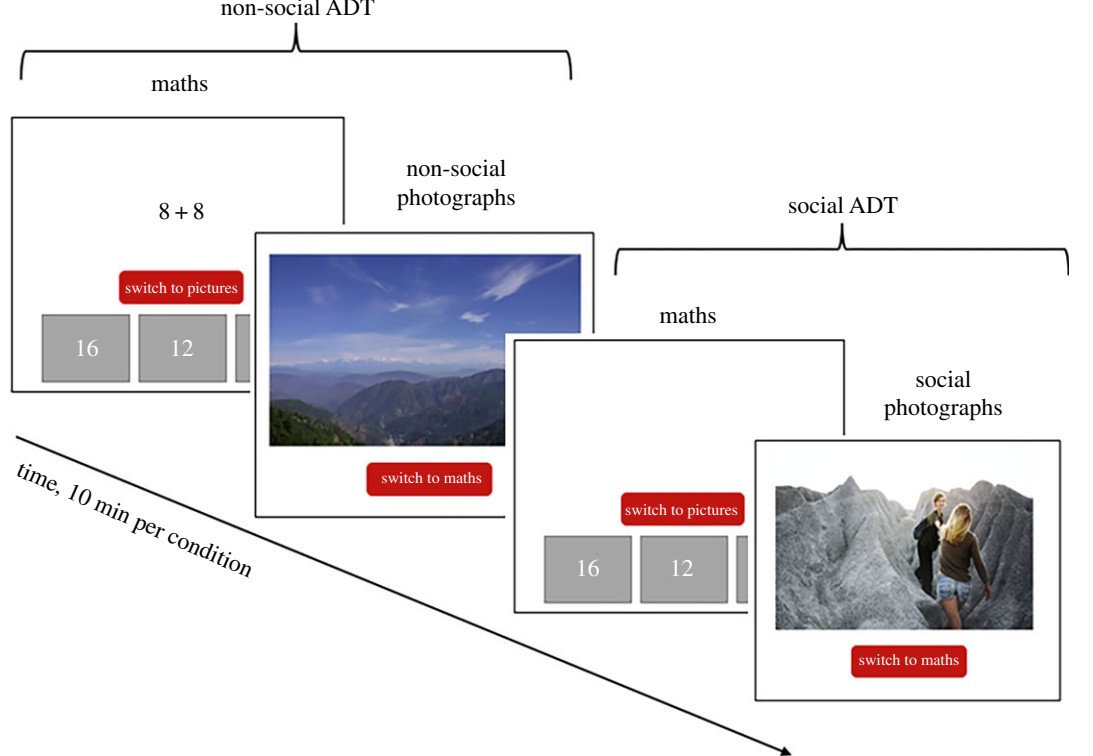

**Figure 1.** Task schematic.

Participants were initially presented with an introduction screen that emphasized the benefits of practising simple maths problems: 'Research has shown that completing simple maths questions such as these will help improve your problem-solving skills in the future'. This statement was further emphasized by the experimenter. Participants were then told that at any moment they could switch from the maths and look at some pictures of people (or landscapes). They were told that they could switch back and forth between the maths and pictures as many times as they liked within the 10 min block. Note that the task length was 10 min for all participants, regardless of the activity they chose. Before the task began, each participant was shown examples of the type of maths problems they would encounter, and the type of pictures they could view. The task always began with the maths questions. For each condition we were able to compute the percentage of time each individual spent doing the maths questions compared to looking at the photographs. Time looking at pictures was automatically recorded in ms by the program, and defined as the time between the button press to exit maths and look at pictures to the button press to exit the pictures and do maths.

After each version of the ADT, participants were asked to rate how enjoyable they found the maths and the show-reel of photographs, on a scale from 0 (not enjoyable) to 100 (extremely enjoyable), unless they spent 100% of their time on the maths, in which case they did not have to provide a rating for the photographs.

Once the participant had read the instructions the experimenter clarified any questions and left the participant to complete the ADT on their own in the testing room. The experimenter was not present in the room during the task in order to minimize any demand characteristics that may arise due to the participant believing that they should spend more time doing maths as the experimenter was watching.

## 2.3. Statistical analysis

Data were analysed using SPSS 24 (IBM Corp., Armonk, NY). In order to test our first hypothesis, a 2 (age group: adult, adolescent) × 2 (condition: social, non-social) mixed model repeated-measures ANOVA was performed to investigate the effect of age and task condition on diligence scores (i.e. time spent doing maths). To test our second hypothesis, we performed two further 2 (age group) × 2 (condition) mixed model repeated-measures ANOVAs on the enjoyment ratings, for the maths questions (see electronic supplementary material) and the photographs. Pairwise $t$-tests were performed to interrogate simple effects, and were Bonferroni corrected for multiple comparisons. For each ANOVA, we then ran follow-up ANCOVAs controlling for autistic traits, social reward and grit.

Pearson's $r$ was used to test our third hypothesis. We investigated the correlation between the participants' picture rating scores and the time spent looking at the pictures, for each condition. We first performed two correlations, one for each condition, including all participants. We then split the analysis by age group, and compared the correlations between the two groups using Fisher's Z. The data is available here [23].

# 3. Results

## 3.1. Percentage of time spent looking at photographs in the ADT

There was no main effect of age group ($F_{1,89} = 2.146$, $p = 0.146$, $\eta^2 = 0.024$) or condition ($F_{1,89} = 0.135$, $p = 0.714$, $\eta^2 = 0.001$) on the percentage of time spent looking at the photographs. However, there was a significant interaction between age group and condition ($F_{1,89} = 7.090$, $p = 0.009$, $\eta^2 = 0.074$) (figure 2; see table 1 for descriptives). This was driven by adolescents spending less time looking at the non-social photographs, compared with adults ($t(89) = -2.867$, $p = 0.02$). There was no significant difference in the time spent looking at the social photographs between adolescents and adults ($t(89) = -0.527$, $p = 1.00$). Adolescents spent more time looking at the social photographs compared to the non-social photographs ($t(44) = -2.197$, $p = 0.033$); however, this finding did not survive Bonferroni correction for multiple comparisons (correction for four *post hoc* $t$-tests; $p = 0.132$). There was no significant difference between the time spent looking at the social photographs compared to the non-social photographs for the adults ($t(45) = 1.586$, $p = 0.48$). When controlling for autism-spectrum quotient (AQ), there was a significant interaction between age group and condition ($F_{1,86} = 6.187$, $p = 0.015$, $\eta^2 = 0.067$). Main effects of age and condition were non-significant ($p > 0.05$). Similarly, when controlling for social reward and grit there was also a significant interaction between age group and condition ($p < 0.05$) and all other main effects were non-significant ($p > 0.05$). The results did not change when the data were analysed continuously, controlling for autistic traits, social reward and grit.

One adult participant answered no maths questions in either condition and another adult answered no maths questions in the non-social condition. Participants' accuracy (proportion of correct answers) on the maths task was very high for both the adolescents and adults in each condition. Within the social condition, accuracy on maths was 95.73% (s.d. 5.06) for adolescents and 97.78% (s.d. 4.77) for adults ($t(88) = 1.98$, $p = 0.051$). In the non-social condition, accuracy on maths was 96.79% (s.d. 3.18) for adolescents and 98.25% (s.d. 3.45) for adults ($t(88) = 2.06$, $p = 0.04$). The reaction time data showed that adolescents took longer to choose an answer to the maths questions compared to the adults, in both the social ($t(88) = 5.99$, $p = 0.001$) and non-social ($t(87) = 4.66$, $p = 0.001$) condition.

## 3.2. Photograph enjoyment ratings following the ADT

18 of the 45 adolescent participants and 8 of the 46 adult participants were not included in this analysis as they chose to spend all their time answering maths questions in either the social or non-social condition or both. There was no significant main effect of age group ($F_{1,63} = 1.966$, $p = 1.66$, $\eta^2 = 0.030$) or condition ($F_{1,63} = 0.564$, $p = 0.455$, $\eta^2 = 0.009$) on the picture enjoyment ratings. However, there was a significant interaction between age group and condition ($F_{1,63} = 12.40$, $p = 0.001$, $\eta^2 = 0.164$). Pairwise comparisons showed that there was a significant difference between the enjoyment ratings for adults compared with adolescents for the non-social pictures, such that adults showed a greater preference for the non-social photographs ($t(63) = -3.156$, $p = 0.008$), but not for the social pictures ($t(63) = 0.653$, $p = 1.00$). Adolescents reported enjoying the social pictures significantly more than the non-social pictures ($t(26) = 2.705$, $p = 0.048$), while adults reported enjoying the non-social pictures significantly more than the social pictures ($t(37) = 2.814$, $p = 0.032$) (figure 3; see electronic supplementary material, table S1 for descriptives). When controlling for AQ, there was a significant interaction between age group and condition ($F_{1,62} = 10.04$, $p = 0.002$, $\eta^2 = 0.137$). Main effects of age and condition were non-significant ($p > 0.05$). Similarly, when controlling for social reward and grit there was also a significant interaction between age group and condition ($p < 0.05$) and all other main effects were non-significant ($p > 0.05$).

## 3.3. Correlations

Across all participants, there was a significant positive correlation between the time spent looking at the social pictures and the enjoyment ratings provided for the social pictures ($r(73) = 0.274$, $p = 0.017$)

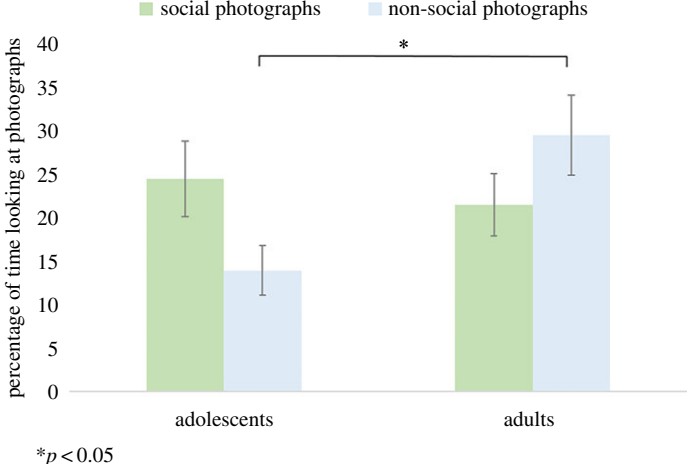

**Figure 2.** Mean percentage of time spent looking at the photographs in each condition, for adolescents and adults. Asterisks indicate Bonferroni corrected *p*-values.

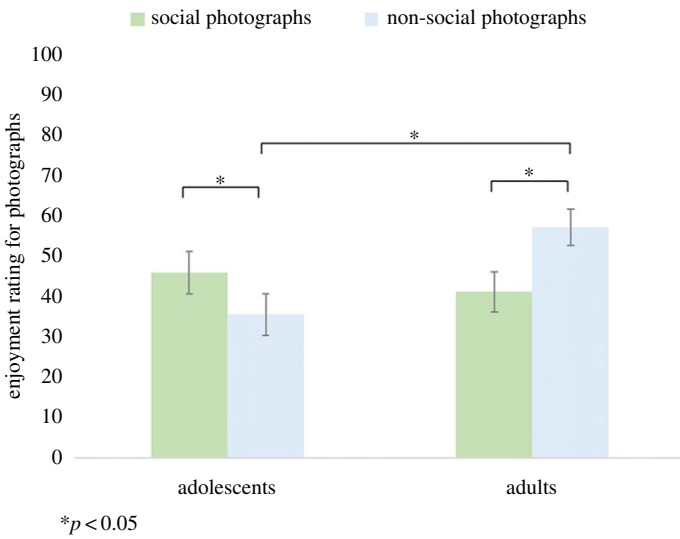

**Figure 3.** Mean enjoyment ratings for the photographs in each condition, for adolescents and adults. Asterisks indicate Bonferroni corrected *p*-values.

**Table 1.** Means and standard error for the time spent looking at the photographs and the self-reported enjoyment ratings for the photographs, in each condition of the ADT.

|  | adolescents | | adults | |
|---|---|---|---|---|
|  | social ADT | non-social ADT | social ADT | non-social ADT |
| % time looking at photographs | $M = 24.4\%$ | $M = 13.87\%$ | $M = 21.43\%$ | $M = 29.41\%$ |
|  | s.e. = 4.35 | s.e. = 2.86 | s.e. = 3.59 | s.e. = 4.57 |
|  | $N = 45$ | $N = 45$ | $N = 46$ | $N = 46$ |
| enjoyment ratings for photographs | $M = 45.89$ | $M = 35.48$ | $M = 41.11$ | $M = 57.16$ |
|  | s.e. = 5.2 | s.e. = 5.08 | s.e. = 4.94 | s.e. = 4.52 |
|  | $N = 27$ | $N = 27$ | $N = 38$ | $N = 38$ |

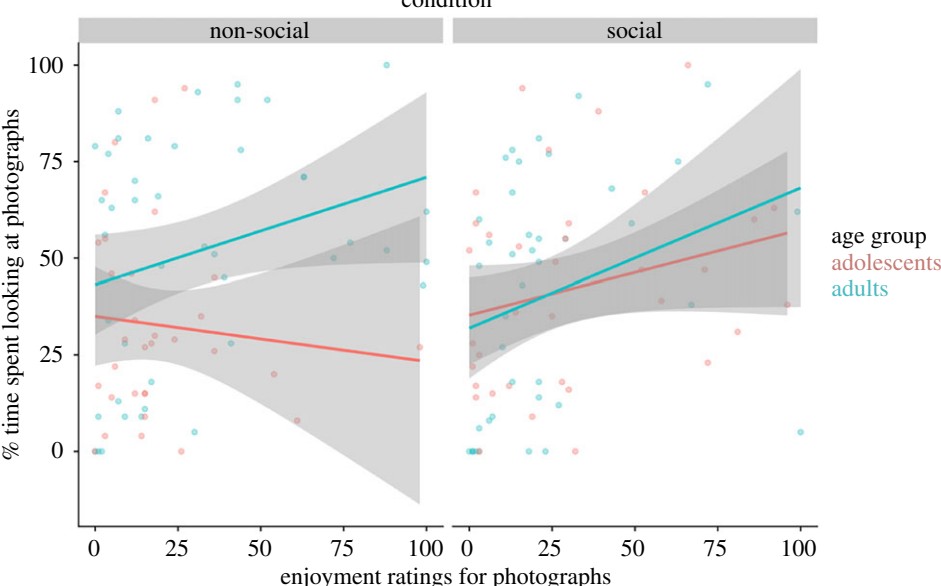

**Figure 4.** The relationship between the enjoyment ratings and time spent looking at the photographs, in the non-social ADT and the social ADT.

(figure 4). There was also a significant positive correlation between the time spent looking at the non-social pictures in the non-social ADT and the enjoyment ratings provided for the non-social pictures ($r(73) = 0.232$, $p = 0.045$) (figure 4). When splitting the data by age group, within the non-social ADT, there was non-significant correlation between the time spent looking at the non-social pictures and the enjoyment ratings provided for the non-social pictures for adolescents ($r(30) = -0.095$, $p = 0.606$). There was also a non-significant correlation for the adults ($r(41) = 0.286$, $p = 0.063$). There was a significant difference between these two correlations, with the adults showing a stronger correlation between enjoyment ratings for and time spent looking at non-social photographs than the adolescents (Fisher's $z = 2.66$, $p = 0.008$). Within the social ADT, there was a non-significant correlation between the time spent looking at the social pictures and the enjoyment ratings provided for the social pictures by the adolescents ($r(33) = 0.253$, $p = 0.143$). There was also a non-significant correlation for the adults ($r(38) = 0.297$, $p = 0.063$). There was no significant difference between these two correlations (Fisher's $z = -0.2$, $p = 0.84$).

## 4. Discussion

The current study investigated the degree to which the presence of social versus non-social stimuli affects academic diligence in female adolescents and adults. In line with our second hypothesis, adolescents reported enjoying the social photographs significantly more than the non-social photographs, while the adults reported enjoying the non-social photographs significantly more than the social photographs. There was no significant difference in the time spent looking at the social photographs between the adolescents and adults. However, adults spent significantly more time than adolescents looking at the non-social photographs, suggesting that adolescents were less motivated to look at the non-social stimuli in comparison to the adults. The correlation between self-reported enjoyment of the pictures and task performance was stronger for adults than for adolescents in non-social condition, suggesting that explicit enjoyment ratings are a stronger predictor of choice behaviour in adults than in adolescents.

Our results showed that adolescents spent less time looking at the non-social photographs, compared to the adults, in the non-social ADT; in other words, adolescents were more diligent that adults in the non-social condition of the ADT. Thus, adolescents showed a reduced motivation to look at non-social stimuli compared with adults. While this is broadly in line with our first hypothesis, we found no difference between the time the two groups spent looking at the social photographs in the social ADT, which does not support our first hypothesis.

In line with our second hypothesis, adolescents reported enjoying the social photographs more than the non-social photographs, while the opposite was true for the adults, who reported enjoying the

non-social photographs more than the social photographs. In addition, and consistent with the ADT-preference data (photograph ratings), adolescents reported enjoying the non-social pictures less than did adults. In contrast, adolescents and adults did not differ in the degree to which they enjoyed looking at the social pictures. The photograph enjoyment ratings thus mirror the behavioural finding that adolescents spent less time looking at the non-social photographs than did adults, in the non-social ADT. Our findings demonstrate that adolescents self-reported a preference for social over non-social stimuli, while adults explicitly reported a preference for the non-social over social stimuli. Taken together with the ADT-preference findings, these data suggest that between adolescence and adulthood, preference switches from social (people) to non-social (landscapes) stimuli.

For both the social and non-social ADT, there was no significant correlation between time spent looking at the pictures and reported enjoyment of the pictures for adolescents, and only a trend for adults. Thus, in both groups, explicit enjoyment ratings were not a strong predictor of choice on the ADT. However, there was a significant difference between the correlations for the two age groups within the non-social ADT, which was driven by a stronger relationship between self-reported preference and choice behaviour in adults compared with adolescents. In non-social ADT, adults' choice behaviour more closely resembled their self-reported preferences, while the behaviour of adolescents was less related to their self-reported preferences. These results are consistent with previous findings that metacognitive and introspective abilities—the capacity to reflect on one's own thoughts or behaviours—continue to develop during adolescence [24]. For example, in one study of 11- to 41-year-olds, the relationship between performance on a visual task and participants' confidence in their performance increased during adolescence, before plateauing into adulthood [24]. Our findings reveal a similar trend, whereby the self-reported preferences, reported after each version of the ADT, better reflected choice behaviour during the non-social ADT for adults, compared with adolescents.

Previous studies involving adolescent participants have also noted the absence of any significant relationships between self-reported 'liking' and behaviour. For example, in one previous study that investigated social motivation in adolescence, a social incentive delay task was carried out by a group of 8- to 16-year-olds, where social rewards were operationalized as smiling faces and verbal approval. Although the subjective value of the faces increased with the intensity of the faces' happiness, this intensity effect was not reflected in behaviour during a social incentive delay task [25]. Therefore, our results add to a broader literature that distinguishes self-reported liking from behaviour during adolescence, which we interpret within the context of developing introspective awareness during adolescence.

# 5. Limitations

There were a number of limitations of the current study. Only female participants were included, in order to increase sample homogeneity; however, this prevents any comparisons between genders. One limitation of our experimental design was that participants who chose not to look at any pictures (and spent all their time on the maths task), and vice versa, did not provide an enjoyment rating. This reduced the number of individuals included in the analyses of the enjoyment ratings. In future versions of the task, requiring all participants to trial the pictures and maths would provide a complete set of ratings. Given that 18 of the 45 adolescent participants and 8 of the 46 adult participants chose to spend all their time on the maths in at least one of the conditions, the strength of the stimuli (i.e. passively viewing photographs) may have been too weak to elicit a distraction effect for these individuals. Future studies of this nature should investigate the use of a social versus non-social distraction task, such as engaging in a virtual game with other players (social) versus a single player game (non-social). Nevertheless, previous work using the original ADT paradigm in a sample of 40 adolescent girls aged 14–15 years found the average percentage time spent on the maths was 84% [22], comparable to the percentages observed in this study.

Another consideration is that the way in which our task conceptualizes social preference differs to some degree to previous tasks aimed to elicit preference behaviour. Previous tasks such as those discussed in the introduction [13,14,16] closely resemble a positive reinforcement paradigm, whereby individuals receive a positive reward following an action, e.g. a button press. In our task, individuals' preference behaviour was assessed within a context of academic diligence, closely resembling a negative reinforcement paradigm, whereby individuals cease a 'boring' task in order to engage with something more appealing (the pictures). The pictures therefore act as a potential distractor, and as

such we interpret this distraction effect as the extent to which individuals exhibit a preference towards the social versus non-social stimuli.

# 6. Conclusion

The present study assessed the degree to which adolescents and adults show a preference for social versus non-social stimuli, within the context of an academic diligence task. We found an age-related effect on task performance, whereby adolescents chose to view the non-social stimuli significantly less than did the adults in the non-social ADT, suggesting that adolescents are less motivated to seek out non-social stimuli than are adults. Adolescents reported enjoying the social stimuli more than the non-social stimuli, while the reverse was true for adults. Taken together, our results suggest that the value of non-social, relative to social, stimuli increases with age. Finally, adolescents' self-reported preferences showed a less strong significant relationship with behaviour in the non-social ADT, than did adults' self-reported preferences, which more closely related to their choice behaviour in the non-social ADT. These results support the notion that metacognitive and introspective abilities are still developing during adolescence.

Ethics. The study followed the Research Ethics Guidelines and was approved by University College London Research Ethics Committee (3453/001). Informed consent was obtained from all participants over the age of 18 and from parents/guardians for individuals under 18 years of age.

Data accessibility. Data available at https://doi.org/10.17605/OSF.IO/XVDJK.

Authors' contributions. J.L.A. was involved in the design, data collection, data analysis and writing. L.F. was involved in the design and writing. C.G. was involved in data collection. S.J.B. was involved in the design, data analysis and writing. All authors gave final approval for publication.

Competing interests. We declare no competing interests.

Funding. J.L.A. is funded by a Medical Research Council Studentship. L.F. and C.G. are funded by a Wellcome Trust grant. S.J.B. is funded by Wellcome, the Jacobs Foundation and GCRF.

Acknowledgements. Thanks to Saz Ahmed for providing helpful comments on an earlier version of the manuscript.

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
