## [Reviewer comments · Royal Society Open Science]

Review History

RSOS-190165.R0 (Original submission)

Review form: Reviewer 1

Is the manuscript scientifically sound in its present form?

Yes

Are the interpretations and conclusions justified by the results?

Yes

Is the language acceptable?

Yes

Is it clear how to access all supporting data?

Not Applicable

Do you have any ethical concerns with this paper?

No

Have you any concerns about statistical analyses in this paper?

No

Recommendation?

Accept with minor revision (please list in comments)

Comments to the Author(s)

The goal of this study was to determine whether adolescents, as compared to adults, exhibit a preference for social versus non-social stimuli, using a paradigm designed to examine academic diligence. The rationale is clear, the paper is well-written and the methods are sound. The results are interesting and confirm assumptions about adolescents' greater motivation for social versus non-social stimuli. Addressing the few issues below could help strengthen the paper.

- 1) How was time spent looking at the pictures assessed?
- 2) Given the hypotheses of the study, it is surprising that 40% of adolescents chose to spend all their time answering maths questions. Did looking at the photos increase time on the task? From the procedures descriptions, it does not appear to be the case but please confirm.
- 3) Table 1. Why are there fewer number of ratings for the enjoyment for both groups (n=27 in adolescents; n=38 in adults)?
- 4) Are there RT data on the enjoyment ratings?
- 5) Are there RT data on the maths problems?
- 6) How many maths problems did the adolescents complete on average before switching to the pictures? Did this number differ from adults?
- 7) Does the behavior relate to the WASI matrix reasoning subscale score or was this controlled for in analyses?
- 8) How accurate was performance on the maths problems?

Review form: Reviewer 2 (Indu Dubey)**Is the manuscript scientifically sound in its present form?**

No

Are the interpretations and conclusions justified by the results?

No

Is the language acceptable?

Yes

Recommendation?

Reject

Comments to the Author(s)

The study aims to compare academic diligence in the context of social and non-social distractors in two age groups using an ADT task. The authors report that "the current study was developed to explore whether adolescents, compared to adults, show a preference for social versus non-social stimuli" (page 4) as they discuss the study in relation to previous studies like Dubey et al (2015), Silva et al (2015), and Ewing et al (2013). However, the way previous studies conceptualise "social preference" is different from the present study. All these three studies (Dubey et al, Silva et al, and Ewing et al) conceptualise social preference as a "positive reward" i.e. a positive consequence in response to an action such as button press. However, in the present study, the

conceptualisation of “social preference” is close to “negative reinforcement” i.e. termination of an action (here the maths task) resulting in a positive state (of watching images). In that sense, the present task is not a comparable measure of the “preference for social or non-social stimuli” to the previous studies. This difference can also help to explain the results that show that none of the groups had a significant difference in the time they spent on looking at the social or non-social stimuli. This is because perhaps both the stimuli served as equally “effective” negative reinforcement for a boring task. Therefore, in my impression, the task used in the present study cannot be taken as a direct measure of the “preference for social versus non-social stimuli” as suggested by the authors. I would be happy to know the author’s take on this comment.

Another major limitation of the study is that the groups being compared are not matched on some very important variables. Since social/non-social preference can have a strong relation with autistic traits authors have collected data on AQ trait measure and social reward questionnaire for the participants. The results from these tasks presented in the supporting information have a high significance when interpreting the main results of the task. The two age groups have a significant difference in their autistic traits, with adults scoring significantly higher than adolescents. Similarly, adults have a significantly lower score on most domains of social reward questionnaire than adolescents. The higher autistic traits in adult participants can explain the significantly higher rating for non-social stimuli in this group, however, this finding has not been discussed anywhere in the study. To test any hypothesis and make claims based on the age difference between the groups, authors would need to control for the effect of autistic traits.

Nearly 40% of the adolescents did not give rating for social/non-social stimuli as they preferred doing the maths 100% of the time. This leaves with a much smaller number of participants in the adolescent group to be compared to the adult group for the rating of the stimuli.

Authors present an inaccurate interpretation of the correlation results “The correlation between self-reported enjoyment of the pictures and task performance was stronger for adults than for adolescents, in both the social and non-social conditions, suggesting that explicit enjoyment ratings are a stronger predictor of choice behaviour in adults than in adolescents.” (Page 11-12). The comparison of correlations for the two groups shows a significant difference only for the non-social condition and not for the social condition.

There are some minor mistakes that the authors might want to correct.
Citation style on page 13 line 30 does not match the rest of the document.
Spelling mistake in line 29 page 4 for word “Neutral”

Review form: Reviewer 3 (Daniel Horn)

Is the manuscript scientifically sound in its present form?

Yes

Are the interpretations and conclusions justified by the results?

Yes

Is the language acceptable?

Yes

Is it clear how to access all supporting data?

Yes

Do you have any ethical concerns with this paper?

No

Have you any concerns about statistical analyses in this paper?

No

Recommendation?

Accept with minor revision (please list in comments)

Comments to the Author(s)

This is a fine piece of scholarship. Coming from a different discipline, admittedly, I admire its simplicity and straightforwardness.

I have one major and more minor questions/suggestions to make.

1) Problem: the stimuli vs. the task

a. The strength of the stimuli

In the initial ADT (Galla et al. 2014) (high school graduates) spent about half their time (54%) on the diligence task. In this study, this ratio is much higher, in between 70% to 84%, depending on the condition and the age group. Moreover, in the current study, 40% of adolescents and 17% of adults spent all their time on the math tasks. Don't you think that the strength of the stimuli was too weak? In my opinion, if 40% of all adolescents spend their time on a tedious math task, the alternative options were either too boring, or you have spent too much effort in convincing them that solving simple math tasks is good for them.

b. The differing incentives for adults and adolescents

All in all, I am not convinced that adults had the same incentives to do the diligence tasks as adolescents. While solving tedious but easy math problems might benefit adolescents in their school life, it is less convincing that the same applies to adults. This is, I believe, the reason for the different ratio of adults vs. adolescents, who spent all their time on the diligence task.

And from this comes a more problematic feature: you don't have enjoyment ratings for those, who did not spend time looking at pictures. Thus for 40% of adolescents and for 17% of adults you have missing values for the enjoyment ratings. If we assume that those, who did not look at pictures, would have scored lower on the enjoyment scale (a harsh but rational assumption), the enjoyment ratings for both groups would drop, but it would drop much more for adolescents than for adults, meaning that the gap in the enjoyment rating for the non-social ADT would be even larger. (Which would strengthen your expected results.)

2) Minor suggestions/questions:

a. Also, I would very much like to see the distribution of the time-spent on diligence tasks under the two conditions for both age groups. Are these distributions similar in shape? Or do children cluster much more on the two ends, while adults distribute themselves more equally? Or?

b. According to table 1, the same amount of people spent all their time on the diligence tasks under both conditions? Is it correct? If yes - which I would find curious - who are these people, who do not look at any of the pictures, not even once? Could you report some background characteristics on them? As the ratio of these is different between the age groups, this might be very important for your results.

c. While the division of the age-groups (adults vs. adolescents) makes the analysis much simpler, I am not convinced that the difference between ages 17 and 23 (oldest adolescent vs. youngest adult), is much larger than between ages 11 and 17 (both adolescents) or between 23 and 33 (both adults). Instead of an ANOVA analysis, you could also run simple regressions using age as a continuous measure and use the different conditions as controls and as interaction effects. I would assume that the distraction effect from the diligence task towards the stimuli should be negatively sloping for the social stimulus by age. (And also: more control variables could and should be included, such as the WASI or the social reward and grit scales, to see how much these

mediate your results. Using multivariate framework – regressions – you could also test functional form differences – e.g. linear age effects vs. quadratic age effect.)

d. Also, in the future, you might want to consider “forcing” all participants to look at pictures as well, so that you could get enjoyment ratings for them too.

All in all, this is a fine piece of scholarship, which I suggest to be published, with the modifications suggested above.

Decision letter (RSOS-190165.R0)

17-Jul-2019

Dear Mr Andrews,

The editors assigned to your paper ("The effect of social preference on academic diligence in adolescence") have now received comments from reviewers. We would like you to revise your paper in accordance with the referee and Associate Editor suggestions which can be found below (not including confidential reports to the Editor). Please note this decision does not guarantee eventual acceptance.

Please submit a copy of your revised paper before 09-Aug-2019. Please note that the revision deadline will expire at 00.00am on this date. If we do not hear from you within this time then it will be assumed that the paper has been withdrawn. In exceptional circumstances, extensions may be possible if agreed with the Editorial Office in advance. We do not allow multiple rounds of revision so we urge you to make every effort to fully address all of the comments at this stage. If deemed necessary by the Editors, your manuscript will be sent back to one or more of the original reviewers for assessment. If the original reviewers are not available, we may invite new reviewers.

- Data accessibility

<http://datadryad.org/submit?journalID=RSOS&manu=RSOS-190165>

- Competing interests

- Authors' contributions

- Acknowledgements

- Funding statement

Kind regards,

on behalf of Dr Anastasia Christakou (Associate Editor) and Essi Viding (Subject Editor)
 openscience@royalsociety.org

Reviewers' Comments to Author:

Reviewer: 1

Comments to the Author(s)

The goal of this study was to determine whether adolescents, as compared to adults, exhibit a preference for social versus non-social stimuli, using a paradigm designed to examine academic diligence. The rationale is clear, the paper is well-written and the methods are sound. The results are interesting and confirm assumptions about adolescents' greater motivation for social versus non-social stimuli. Addressing the few issues below could help strengthen the paper.

- 1) How was time spent looking at the pictures assessed?
- 2) Given the hypotheses of the study, it is surprising that 40% of adolescents chose to spend all their time answering maths questions. Did looking at the photos increase time on the task? From the procedures descriptions, it does not appear to be the case but please confirm.
- 3) Table 1. Why are there fewer number of ratings for the enjoyment for both groups (n=27 in adolescents; n=38 in adults)?
- 4) Are there RT data on the enjoyment ratings?
- 5) Are there RT data on the maths problems?
- 6) How many maths problems did the adolescents complete on average before switching to the pictures? Did this number differ from adults?
- 7) Does the behavior relate to the WASI matrix reasoning subscale score or was this controlled for in analyses?
- 8) How accurate was performance on the maths problems?

Reviewer: 2

Comments to the Author(s)

The study aims to compare academic diligence in the context of social and non-social distractors in two age groups using an ADT task. The authors report that "the current study was developed to explore whether adolescents, compared to adults, show a preference for social versus non-social stimuli" (page 4) as they discuss the study in relation to previous studies like Dubey et al (2015), Silva et al (2015), and Ewing et al (2013). However, the way previous studies conceptualise "social preference" is different from the present study. All these three studies (Dubey et al, Silva et al, and Ewing et al) conceptualise social preference as a "positive reward" i.e. a positive consequence in response to an action such as button press. However, in the present study, the conceptualisation of "social preference" is close to "negative reinforcement" i.e. termination of an action (here the maths task) resulting in a positive state (of watching images). In that sense, the present task is not a comparable measure of the "preference for social or non-social stimuli" to the previous studies. This difference can also help to explain the results that show that none of the groups had a significant difference in the time they spent on looking at the social or non-social stimuli. This is because perhaps both the stimuli served as equally "effective" negative reinforcement for a boring task. Therefore, in my impression, the task used in the present study cannot be taken as a direct measure of the "preference for social versus non-social stimuli" as suggested by the authors. I would be happy to know the author's take on this comment.

Another major limitation of the study is that the groups being compared are not matched on

some very important variables. Since social/non-social preference can have a strong relation with autistic traits authors have collected data on AQ trait measure and social reward questionnaire for the participants. The results from these tasks presented in the supporting information have a high significance when interpreting the main results of the task. The two age groups have a significant difference in their autistic traits, with adults scoring significantly higher than adolescents. Similarly, adults have a significantly lower score on most domains of social reward questionnaire than adolescents. The higher autistic traits in adult participants can explain the significantly higher rating for non-social stimuli in this group, however, this finding has not been discussed anywhere in the study. To test any hypothesis and make claims based on the age difference between the groups, authors would need to control for the effect of autistic traits.

Nearly 40% of the adolescents did not give rating for social/non-social stimuli as they preferred doing the maths 100% of the time. This leaves with a much smaller number of participants in the adolescent group to be compared to the adult group for the rating of the stimuli.

Authors present an inaccurate interpretation of the correlation results "The correlation between self-reported enjoyment of the pictures and task performance was stronger for adults than for adolescents, in both the social and non-social conditions, suggesting that explicit enjoyment ratings are a stronger predictor of choice behaviour in adults than in adolescents." (Page 11-12). The comparison of correlations for the two groups shows a significant difference only for the non-social condition and not for the social condition.

There are some minor mistakes that the authors might want to correct.
Citation style on page 13 line 30 does not match the rest of the document.
Spelling mistake in line 29 page 4 for word "Neutral"

Reviewer: 3

Comments to the Author(s)

This is a fine piece of scholarship. Coming from a different discipline, admittedly, I admire its simplicity and straightforwardness.

I have one major and more minor questions/suggestions to make.

1) Problem: the stimuli vs. the task

a. The strength of the stimuli

In the initial ADT (Galla et al. 2014) (high school graduates) spent about half their time (54%) on the diligence task. In this study, this ratio is much higher, in between 70% to 84%, depending on the condition and the age group. Moreover, in the current study, 40% of adolescents and 17% of adults spent all their time on the math tasks. Don't you think that the strength of the stimuli was too weak? In my opinion, if 40% of all adolescents spend their time on a tedious math task, the alternative options were either too boring, or you have spent too much effort in convincing them that solving simple math tasks is good for them.

b. The differing incentives for adults and adolescents

All in all, I am not convinced that adults had the same incentives to do the diligence tasks as adolescents. While solving tedious but easy math problems might benefit adolescents in their school life, it is less convincing that the same applies to adults. This is, I believe, the reason for the different ratio of adults vs. adolescents, who spent all their time on the diligence task.

And from this comes a more problematic feature: you don't have enjoyment ratings for those, who did not spend time looking at pictures. Thus for 40% of adolescents and for 17% of adults you have missing values for the enjoyment ratings. If we assume that those, who did not look at pictures, would have scored lower on the enjoyment scale (a harsh but rational assumption), the enjoyment ratings for both groups would drop, but it would drop much more for adolescents

than for adults, meaning that the gap in the enjoyment rating for the non-social ADT would be even larger. (Which would strengthen your expected results.)

2) Minor suggestions/questions:

a. Also, I would very much like to see the distribution of the time-spent on diligence tasks under the two conditions for both age groups. Are these distributions similar in shape? Or do children cluster much more on the two ends, while adults distribute themselves more equally? Or?

b. According to table 1, the same amount of people spent all their time on the diligence tasks under both conditions? Is it correct? If yes – which I would find curious – who are these people, who do not look at any of the pictures, not even once? Could you report some background characteristics on them? As the ratio of these is different between the age groups, this might be very important for your results.

c. While the division of the age-groups (adults vs. adolescents) makes the analysis much simpler, I am not convinced that the difference between ages 17 and 23 (oldest adolescent vs. youngest adult), is much larger than between ages 11 and 17 (both adolescents) or between 23 and 33 (both adults). Instead of an ANOVA analysis, you could also run simple regressions using age as a continuous measure and use the different conditions as controls and as interaction effects. I would assume that the distraction effect from the diligence task towards the stimuli should be negatively sloping for the social stimulus by age. (And also: more control variables could and should be included, such as the WASI or the social reward and grit scales, to see how much these mediate your results. Using multivariate framework – regressions – you could also test functional form differences – e.g. linear age effects vs. quadratic age effect.)

d. Also, in the future, you might want to consider “forcing” all participants to look at pictures as well, so that you could get enjoyment ratings for them too.

All in all, this is a fine piece of scholarship, which I suggest to be published, with the modifications suggested above.

Author's Response to Decision Letter for (RSOS-190165.R0)

See Appendix A.

Decision letter (RSOS-190165.R1)

09-Aug-2019

Dear Mr Andrews,

I am pleased to inform you that your manuscript entitled "The effect of social preference on academic diligence in adolescence" is now accepted for publication in Royal Society Open Science.

Royal Society Open Science operates under a continuous publication model

(<http://bit.ly/cpFAQ>). Your article will be published straight into the next open issue and this will be the final version of the paper. As such, it can be cited immediately by other researchers. As the issue version of your paper will be the only version to be published I would advise you to check your proofs thoroughly as changes cannot be made once the paper is published.

Kind regards,

on behalf of Dr Anastasia Christakou (Associate Editor) and Essi Viding (Subject Editor)
openscience@royalsociety.org

Appendix A

Response to reviewers (07/08/2019)

Reviewer: 1

Comments to the Author(s)

The goal of this study was to determine whether adolescents, as compared to adults, exhibit a preference for social versus non-social stimuli, using a paradigm designed to examine academic diligence. The rationale is clear, the paper is well-written and the methods are sound. The results are interesting and confirm assumptions about adolescents' greater motivation for social versus non-social stimuli. Addressing the few issues below could help strengthen the paper.

1) How was time spent looking at the pictures assessed?

Addition to manuscript (page 8):

“Time looking at pictures was automatically recorded in ms by the programme, and defined as the time between the button press to exit maths and look at pictures to the button press to exit the pictures and do maths.”

2) Given the hypotheses of the study, it is surprising that 40% of adolescents chose to spend all their time answering maths questions. Did looking at the photos increase time on the task? From the procedures descriptions, it does not appear to be the case but please confirm.

Only 5/46 adolescents spent all their time doing the maths in both conditions; 18/46 of the adolescents chose to spend all their time doing the maths in one condition or more. We were not surprised by this finding, given that in a previous study from our lab employing the traditional ADT (maths vs. Tetris), the mean proportion of time spent on maths was 84.14%, with individual scores ranging from 34.44%-96.67%.

Ref: Fuhrmann, D., Schweizer, S., Leung, J., Griffin, C., & Blakemore, S. J. (2019). The neurocognitive correlates of academic diligence in adolescent girls. *Cognitive Neuroscience*, 10(2), 88-99.

Looking at the pictures did not increase time spent on the task: all participants were given precisely 10 minutes to complete the entire task block, regardless of what activity they chose

From the Methods section (page 8):

“They [participants] were told that they could switch back and forth between the maths and pictures as many times as they liked within the 10-min block”.

Addition to manuscript (page 8):

“Note that the task length was 10 minutes for all participants, regardless of the activity they chose”.

3) Table 1. Why are there fewer number of ratings for the enjoyment for both groups (n=27 in adolescents; n=38 in adults)?

This is because participants who did not look at the pictures at all in a block (and instead spent 100% of their time doing maths) did not provide an enjoyment rating for pictures. The ANOVA on enjoyment ratings only included individuals who provided enjoyment ratings in both conditions, and so the number remains lower than the total N.

From the Methods section (page 8):

“After each version of the ADT, participants were asked to rate how enjoyable they found the maths and the show-reel of photographs, on a scale from 0 (not enjoyable) to 100 (extremely enjoyable), unless they spent 100% of their time on the maths, in which case they did not have to provide a rating for the photographs.”

Enjoyment ratings were not the primary question in our task design. However, we have added this to our limitations section and suggest that future studies should ask all participants to rate a selection of pictures prior to the task.

Addition to manuscript (page 15):

“One limitation of our experimental design was that participants who chose not to look at any pictures (and spent all their time on the maths task) and vice versa, did not provide an enjoyment rating. This reduced the number of individuals included in the analyses of the enjoyment ratings. In future versions of the task, requiring all participants to trial the pictures and maths would provide a complete set of ratings.”

4) Are there RT data on the enjoyment ratings?

We do not have RT data on enjoyment ratings.

5) Are there RT data on the maths problems?

We do have these data, which we have now analysed and added to the Results section. The results show that adolescents were slower than adults were when selecting an answer to the maths questions, in both conditions:

Addition to manuscript (page 10/11):

“The reaction time data showed that, compared to adults, adolescents took longer to select an answer to the maths questions, in both the social ($t(88)= 5.985, p= .001$) and non-social ($t(87)= 4.661, p= .001$) condition.”

6) How many maths problems did the adolescents complete on average before switching to the pictures? Did this number differ from adults?

Thank you for this question. We ran a 2 (Condition: Social vs. Non Social) x 2 (Age group: Adolescent vs. Adult) repeated measures ANOVA on this data. We found no main effects of age or condition, or interaction effect.

Addition to manuscript (Supplemental page 2):

	Social Condition		Non-Social Condition	
	Adolescents	Adults	Adolescents	Adults
Mean	51.04	79.39	56.38	72.09
SD	68.72	88.01	70.06	101.6

Table S2: Mean and standard deviations of the number of maths problems completes before the first switch from maths to pictures, by condition.

“There was no main effect of condition ($F(1,89)=0.01$, $p=9.20$, $\eta^2= .00$) or age group ($F(1,89)=2.30$, $p=.133$, $\eta^2= .025$) on the number of maths problems completed prior to the first switch to look at the pictures. There was no interaction between age group or condition ($F(1,89)=0.42$, $p=.52$, $\eta^2= .01$) (see table S2 for descriptive statistics).”

7) Does the behavior relate to the WASI matrix reasoning subscale score or was this controlled for in analyses?

The WASI matrix reasoning scores do not correlate with behaviour in the task in the current study and therefore we do not to include WASI scores in any further analyses. We have, as asked by the other reviewers, conducted follow up analyses, controlling for autistic traits, social reward and grit.

We are reluctant to include the WASI scores as a covariate because we believe the normalisation of this subscale is problematic. Normalised scores for younger people (teenagers) are lower than those for adults. This means that young people (in their teens) with *higher* raw scores than older people (e.g. in their 30s) are given a *lower* normalised IQ score (based on the age norms provided). This is seriously problematic as it inherently biases younger individuals towards having lower normalised IQ scores than adults – indeed, it is almost impossible for a young person to have a higher normalised score than an adult, even if the young person scores very highly. This makes little sense, as there is a very large literature showing that non-verbal reasoning improves with age between adolescence and adulthood (e.g. Sternberg & Rifkin, 1979; Richland, Morrison & Holyoak, 2006), which is contrary to the normalisation data on which this subscale is based. We suspect the normalisation data might not be accurate. We were unaware of this issue prior to data collection for this study.

Given this issue, we have subsequently moved to using different proxy measures for IQ in our studies. In fact, as a lab, we have decided not to include the WASI matrix-reasoning subscale in any of our analyses. Instead, we have developed a measure of relational reasoning, which addresses the above problem and we intend to use this in the future (<https://osf.io/uvteh/>).

8) How accurate was performance on the maths problems?

Accuracy was very high for both adolescents and adults, reaching ceiling effects, showing that the level of maths was very basic, as intended (and comparable to the 96% accuracy reported in the original ADT; Galla et al., 2014). Accuracy was higher for adults than for adolescents, but given the high degree of accuracy , we do not interpret this further.

Addition to manuscript (page 10):

“One adult participant answered no maths questions in either condition and another adult answered no maths questions in the non-social condition. Participants’ accuracy (proportion of correct answers) on the maths task was very high for both the adolescents and adults in each condition. Within the social condition, accuracy on maths was 95.73% (S.D. 5.06) for adolescents and 97.78% (S.D. 4.77) for adults ($t(88)= 1.98$, $p= .051$). In the non-social condition, accuracy on maths was 96.79% (3.18) for adolescents and 98.25% (3.45) for adults ($t(88)= 2.06$, $p= .04$). The reaction time data showed that adolescents took longer to choose an answer to the maths questions compared to the adults, in both the social ($t(88)= 5.99$, $p= .001$) and non-social ($t(87)= 4.66$, $p= .001$) condition. “

Reviewer: 2

Comments to the Author(s)

The study aims to compare academic diligence in the context of social and non-social distractors in two age groups using an ADT task. The authors report that “the current study was developed to explore whether adolescents, compared to adults, show a preference for social versus non-social stimuli” (page 4) as they discuss the study in relation to previous studies like Dubey et al (2015), Silva et al (2015), and Ewing et al (2013). However, the way previous studies conceptualise “social preference” is different from the present study. All these three studies (Dubey et al, Silva et al, and Ewing et al) conceptualise social preference as a “positive reward” i.e. a positive consequence in response to an action such as button press. However, in the present study, the conceptualisation of “social preference” is close to “negative reinforcement” i.e. termination of an action (here the maths task) resulting in a positive state (of watching images). In that sense, the present task is not a comparable measure of the “preference for social or non-social stimuli” to the previous studies. This difference can also help to explain the results that show that none of the groups had a significant difference in the time they spent on looking at the social or non-social stimuli. This is because perhaps both the stimuli served as equally “effective” negative reinforcement for a boring task. Therefore, in my impression, the task used in the present study cannot be taken as a direct measure of the “preference for social versus non-social stimuli” as suggested by the authors. I would be happy to know the author’s take on this comment.

This is a really helpful comment, which we have now reflected on in the Discussion. Whilst we agree that our academic diligence task could be conceptualised as a negative reinforcement task, it was designed to resemble the choices often faced by individuals doing (school) work. We were interested in whether the desire to seek out social stimuli hampered diligence more strongly than non-social stimuli in adolescents compared to adults. We believe this represents a ‘preference’ behaviour, but perhaps the strength with which our stimuli elicit a distraction effect was weak (e.g. in our task individuals passively viewed social vs non-social pictures), which we also now address in the Discussion. Nevertheless, our results show an interaction between age and condition, driven by the non-social stimuli working as a greater distractor for adults than for adolescents. This is also reflected in the enjoyment ratings, which show that adults have a greater preference for non-social stimuli compared to adolescents.

However, this comment is well received and the degree to which our interpretation of social vs non-social preference differs from previous positive reinforcement tasks is now mentioned in the Discussion .

Addition to manuscript (page 15/16):

“Another consideration is that the way in which our task conceptualises social preference differs to some degree to previous tasks aimed to elicit preference behaviour. Previous tasks such as those discussed in the introduction [13, 14, 16] closely resemble a positive reinforcement paradigm, whereby individuals receive a positive reward following an action e.g. a button press. In our task, individuals’ preference behaviour was assessed within a context of academic diligence, closely resembling a negative reinforcement paradigm, whereby individuals cease a ‘boring’ task in order to engage with something more appealing (the pictures). The pictures therefore act as a potential distractor, and as such we interpret this distraction effect as the extent to which individuals exhibit a preference towards the social vs. non-social stimuli.”

Another major limitation of the study is that the groups being compared are not matched on some very important variables. Since social/non-social preference can have a strong relation with autistic traits authors have collected data on AQ trait measure and social reward questionnaire for the participants. The results from these tasks presented in the supporting information have a high significance when interpreting the main results of the task. The two age groups have a significant difference in their autistic traits, with adults scoring significantly higher than adolescents. Similarly, adults have a significantly lower score on most domains of social reward questionnaire than adolescents. The higher autistic traits in adult participants can explain the significantly higher rating for non-social stimuli in this group, however, this finding has not been discussed anywhere in the study. To test any hypothesis and make claims based on the age difference between the groups, authors would need to control for the effect of autistic traits.

Thank you for this comment, which we agree with. We have re-run our analyses on the diligence data controlling for autistic traits, social reward and grit, and the results remain unchanged.

Additions to manuscript:

For the diligence task (page 10):

“When controlling for AQ, there was a significant interaction between age group and condition ($F(1,86)=6.187, p=.015, \eta^2=.067$). Main effects of age and condition were non-significant ($p>.05$). Similarly, when controlling for social reward and grit there was also a significant interaction between age group and condition ($p<.05$) and all other main effects were non-significant ($p>.05$).”

For the picture ratings (page 11):

“When controlling for AQ, there was a significant interaction between age group and condition ($F(1,62)=10.04, p=.002, \eta^2=.137$). Main effects of age and condition were non-significant ($p>.05$). Similarly, when controlling for social reward and grit there was also a significant interaction between age group and condition ($p<.05$) and all other main effects were non-significant ($p>.05$).”

Nearly 40% of the adolescents did not give rating for social/non-social stimuli as they preferred doing the maths 100% of the time. This leaves with a much smaller number of participants in the adolescent group to be compared to the adult group for the rating of the stimuli.

We only included participants who had complete datasets for both the social and non-social condition. Data from 38 of the 46 (82.6%) adults and 27 of the 45 (60%) adolescents data were included in the analysis. As detailed above in response to reviewer 1, we have added to our limitations section the fact we did not ask all participants to provide a rating for the photographs, regardless of the time spent viewing them.

Authors present an inaccurate interpretation of the correlation results “The correlation between self-reported enjoyment of the pictures and task performance was stronger for adults than for adolescents, in both the social and non-social conditions, suggesting that explicit enjoyment ratings are a stronger predictor of choice behaviour in adults than in adolescents.” (Page 11-12). The comparison of correlations for the two groups shows a significant difference only for the non-social condition and not for the social condition.

Thank you for spotting this. We have updated the manuscript in order to rectify this and highlight that the significant difference between correlations was found in the non-social condition, but not the social condition. Minor alterations to wording have been made to the abstract (page 2), the discussion (page 13-14) and conclusion on page 16.

There are some minor mistakes that the authors might want to correct.

Citation style on page 13 line 30 does not match the rest of the document.

Spelling mistake in line 29 page 4 for word “Neutral”

We have corrected these mistakes, thank you for spotting them.

Reviewer: 3

Comments to the Author(s)

This is a fine piece of scholarship. Coming from a different discipline, admittedly, I admire its simplicity and straightforwardness.

I have one major and more minor questions/suggestions to make.

1) Problem: the stimuli vs. the task

a. The strength of the stimuli

In the initial ADT (Galla et al. 2014) (high school graduates) spent about half their time (54%) on the diligence task. In this study, this ratio is much higher, in between 70% to 84%, depending on the condition and the age group. Moreover, in the current study, 40% of adolescents and 17% of adults spent all their time on the math tasks. Don't you think that the strength of the stimuli was too weak? In my opinion, if 40% of all adolescents spend their time on a tedious math task, the alternative options were either too boring, or you have spent too much effort in convincing them that solving simple math tasks is good for them.

Thank you for this comment. Our results are comparable to previous work using the original ADT, for example, in one study the average amount of time spent doing the maths was 84% in a sample of 40 adolescent girls aged 14-15 (Fuhrmann et al., 2019). Therefore, even though we used a novel distractor task (viewing the social vs non-social pictures), overall diligence (percentage of time spent doing the maths vs. looking at pictures) remains less than this previous finding, which used the original ADT design.

However, we acknowledge that, in the original ADT, the alternative task to doing maths was a more interactive option (e.g. playing Tetris), which may have been more appealing/enjoyable to participants than the passive task we used (viewing pictures). In future versions of this task, perhaps having individuals play an alternative interactive social vs. non-social task would better mirror the original ADT – an idea that we now mention also in the limitations section.

Alterations to manuscript (page 15):

“Given that 18 of the 45 adolescent participants and 8 of the 46 adult participants chose to spend all their time on the maths in at least one of the conditions, the strength of the stimuli (i.e. passively viewing photographs) may have been too weak to elicit a distraction effect for these individuals. Future studies of this nature should investigate the use of a social vs. non-social distraction task such as engaging in a virtual game with other players (social) versus a single player game (non-social). Nevertheless, previous work using the original ADT paradigm in a sample of 40 adolescent girls aged 14-15 years found the average percentage time spent on the maths was 84% [22], comparable to the percentages observed in this study.”

Nevertheless, our task still elicited an age by condition interaction, showing that adults, compared to adolescents, spent less time on the maths in the non-social condition, revealing a greater preference for the non-social photographs.

Ref: Fuhrmann, D., Schweizer, S., Leung, J., Griffin, C., & Blakemore, S. J. (2019). The neurocognitive correlates of academic diligence in adolescent girls. *Cognitive Neuroscience*, 10(2), 88-99.

b. The differing incentives for adults and adolescents

All in all, I am not convinced that adults had the same incentives to do the diligence tasks as adolescents. While solving tedious but easy math problems might benefit adolescents in their school life, it is less convincing that the same applies to adults. This is, I believe, the reason for the different ratio of adults vs. adolescents, who spent all their time on the diligence task.

And from this comes a more problematic feature: you don't have enjoyment ratings for those, who did not spend time looking at pictures. Thus for 40% of adolescents and for 17% of adults you have missing values for the enjoyment ratings. If we assume that those, who did not look at pictures, would have scored lower on the enjoyment scale (a harsh but rational assumption), the enjoyment ratings for both groups would drop, but it would drop much more for adolescents than for adults, meaning that the gap in the enjoyment rating for the non-social ADT would be even larger. (Which would strengthen your expected results.)

We understand this point and it speaks to the issue of not asking all participants to provide a rating for both sets of pictures. As we say in our response to the same question from reviewer 1 and 2, we now mention this as a limitation to our study.

Alterations to manuscript (page 15):

“There were a number of limitations of the current study. Only female participants were included in order to increase sample homogeneity, however this prevents any comparisons across gender. One limitation of our experimental design was that participants who chose not to look at any pictures (and spent all their time on the maths task) and vice versa, did not provide an enjoyment rating. This reduced the number of individuals included in the analyses of the enjoyment ratings. In future versions of the task, requiring all participants to trial the pictures and maths would provide a complete set of ratings. Given that 18 of the 45 adolescent participants and 8 of the 46 adult participants chose to spend all their time on the maths in at least one of the conditions, the strength of the stimuli (i.e. passively viewing photographs) may have been too weak to elicit a distraction effect for these individuals. Future studies of this nature should investigate the use of a social vs. non-social distraction task such as engaging in a virtual game with other players (social) vs a single player game (non-social). Nevertheless, previous work using the original ADT paradigm in a sample of 40 adolescent girls aged 14-15, found the average percentage time spent on the maths was 84% [22], comparable to the percentages observed in this study.”

2) Minor suggestions/questions:

a. Also, I would very much like to see the distribution of the time-spent on diligence tasks under the two conditions for both age groups. Are these distributions similar in shape? Or do children cluster much more on the two ends, while adults distribute themselves more equally? Or?

This is a good point, and we now show these data in the supplemental materials. In the social condition the distributions for the two groups are similar, whilst in the non-social a greater number of adolescents choose to spend less time looking at the pictures (and more time on maths). Across each condition and age groups, the distributions appear unimodal.

Additions to manuscript (supplemental material page 5):

Distributions of time spent looking at the pictures

Figure S2: Distribution of time spent looking at non-social pictures in the non-social ADT

Figure S3 Distribution of time spent looking at social pictures in the social ADT

b. According to table 1, the same amount of people spent all their time on the diligence tasks under both conditions? Is it correct? If yes – which I would find curious – who are these people, who do not look at any of the pictures, not even once? Could you report some background characteristics on them? As the ratio of these is different between the age groups, this might be very important for your results.

This was unclear, and in fact these people are not all the same in each condition as we only included participants in the ANOVA who had complete datasets for both the social and non-social condition: 38/46 (82.6%) adults and 27/45 (60%) adolescents provided a picture rating in both the social and non-social condition. We have made an addition to the manuscript on page 11 to make this clearer. Individuals who provided a rating in just one condition were excluded from the analysis. Only five (11%) adolescents and one (2%) adult spent all their time in both conditions on the maths. Background characteristics on these individuals are reported below in relation to the overall average. From this, there appears no notable differences between the background information of these participants and the group means.

	Adolescent mean (N=45)	Adolescent (N=5)	Adult mean (N=46)	Adult (N=1)
Age	15.4	14.6	24.96	24
AQ	15.7	14	19.46	14
SRQ Sociability	5.5	4.8	4.82	6
SRQ Prosocial	6.5	5.96	6.13	6
SRQ Admiration	5.73	5.1	5.79	5
GRIT	2.89	2.63	2.69	2.66

Addition to manuscript (page 11):

“18 of the 45 adolescent participants and 8 of the 46 adult participants were not included in this analysis as they chose to spend all their time answering maths questions in either the social or non-social condition or both.”

c. While the division of the age-groups (adults vs. adolescents) makes the analysis much simpler, I am not convinced that the difference between ages 17 and 23 (oldest adolescent vs. youngest adult), is much larger than between ages 11 and 17 (both adolescents) or between 23 and 33 (both adults). Instead of an ANOVA analysis, you could also run simple regressions using age as a continuous measure and use the different conditions as controls and as interaction effects. I would assume that the distraction effect from the diligence task towards the stimuli should be negatively sloping for the social stimulus by age. (And also: more control variables could and should be included, such as the WASI or the social reward and grit scales, to see how much these mediate your results. Using multivariate framework – regressions – you could also test functional form differences – e.g. linear age effects vs. quadratic age effect.)

Thank you for this suggestion. We have taken this on board and ran the regression analyses suggested on the diligence data, with age as a continuous variable. Given that there was a significant age by condition interaction in the ANOVA, we anticipated the same result would be found when the same data are analysed using regression - it is essentially the same test and the ANOVA has already shown there is a significant difference between the two groups, which will show up as a significant regression. Indeed, omnibus tests revealed the same significant interaction between age and condition (social/non-social) when controlling for autistic traits ($F(1,173) = 4.036, p = .04$). This does not change when we control for grit or social reward scales.

Graph from the regression analysis, with age predicting time spent looking at the pictures, by condition (social/non-social):

Note: Picture time = time spent looking at the pictures, agecon = age (continuous), condition (NS PIC = non-social condition; S PIC = social condition)

Given that these findings do not differ, we have decided to retain the results of the ANOVA/ANCOVA with group as a categorical variable. However, we have updated the manuscript mentioning that the results do not change when age is included in the analysis as a continuous variable. We decided not to compare linear vs quadratic age effects given that we did not collect data for individuals aged 18-22.

Addition to manuscript (page 10)

“When controlling for AQ, there was a significant interaction between age group and condition ($F(1,86)=6.187, p=.015, \eta^2=.067$). Main effects of age and condition were non-significant ($p>.05$). Similarly, when controlling for social reward and grit there was also a significant interaction between age group and condition ($p<.05$) and all other main effects were non-significant ($p>.05$). The results do not change when the data is analysed continuously, controlling for autistic traits, social reward and grit.”

d. Also, in the future, you might want to consider “forcing” all participants to look at pictures as well, so that you could get enjoyment ratings for them too.

This is an excellent point, which we realise would have improved the design of our study. We have added this limitation to our discussion and in future studies of this nature will be sure to design the study in such a way that everyone provides these ratings.

Addition to manuscript (page 15):

“One limitation of our experimental design was that participants who chose not to look at any pictures (and spent all their time on the maths task) and vice versa, did not provide an enjoyment rating. This reduced the number of individuals included in the analyses of the

enjoyment ratings. In future versions of the task, requiring all participants to trial the pictures and maths would provide a complete set of ratings.”

All in all, this is a fine piece of scholarship, which I suggest to be published, with the modifications suggested above.